# The Use of Digital Coronary Phantoms for the Validation of Arterial Geometry Reconstruction and Computation of Virtual FFR

**Giulia Pederzani** [1,2], **Krzysztof Czechowicz** [1,2], **Nada Ghorab** [1,2], **Paul D. Morris** [1,2,3], **Julian P. Gunn** [1,2,3], **Andrew J. Narracott** [1,2], **David Rodney Hose** [1,2] and **Ian Halliday** [1,2,*]

1  Department of Infection, Immunity and Cardiovascular Disease, Faculty of Medicine, Dentistry and Health, University of Sheffield, Sheffield S10 2TN, UK; g.pederzani@sheffield.ac.uk (G.P.); k.czechowicz@sheffield.ac.uk (K.C.); naaghorab1@sheffield.ac.uk (N.G.); paul.morris@sheffield.ac.uk (P.D.M.); j.gunn@sheffield.ac.uk (J.P.G.); a.j.narracott@sheffield.ac.uk (A.J.N.); d.r.hose@sheffield.ac.uk (D.R.H.)
2  INSIGNEO Institute for in silico Medicine, Sheffield S1 3JD, UK
3  Department of Cardiology, Sheffield Teaching Hospitals NHS Foundation Trust, Sheffield S10 2JF, UK
*  Correspondence: i.halliday@sheffield.ac.uk

**Abstract:** We present computational fluid dynamics (CFD) results of virtual fractional flow reserve (vFFR) calculations, performed on reconstructed arterial geometries derived from a digital phantom (DP). The latter provides a convenient and parsimonious description of the main vessels of the left and right coronary arterial trees, which, crucially, is CFD-compatible. Using our DP, we investigate the reconstruction error in what we deem to be the most relevant way—by evaluating the change in the computed value of vFFR, which results from varying (within representative clinical bounds) the selection of the virtual angiogram pair (defined by their viewing angles) used to segment the artery, the eccentricity and severity of the stenosis, and thereby, the CFD simulation's luminal boundary. The DP is used to quantify reconstruction and computed haemodynamic error within the VIRTUheart[TM] software suite. However, our method and the associated digital phantom tool are readily transferable to equivalent, clinically oriented workflows. While we are able to conclude that error within the VIRTUheart[TM] workflow is suitably controlled, the principal outcomes of the work reported here are the demonstration and provision of a practical tool along with an exemplar methodology for evaluating error in a coronary segmentation process.

**Keywords:** digital phantom; coronary angiography; physiology; fractional flow reserve; computational fluid dynamics; percutaneous coronary intervention

## 1. Introduction

Notable progress has been made, since the turn of the century, in the field of medical image enhancement, segmentation, quantification, and registration [1]. While the generation of a three-dimensional (3D) segmentation from a set of medical images is almost commmonplace, the question of its accuracy in representing the patient's health status sufficiently well to inform prognosis and treatment remains one which still requires significant attention. The process of validating the image processing algorithms is therefore of crucial importance. To this end, several options are available: in order of decreasing financial and temporal cost, these are human clinical trials, in vivo animal models, ex vivo human or animal models, physical phantoms and, finally, digital phantoms (DP).

Digital phantoms consist of algorithmically generated data which, in the context of medical imaging, are used to generate synthetic yet realistic images that can be valuable for the calibration of imaging devices, the standardisation of imaging protocols, or the comparison of devices. In addition to cost, the significant advantage of DPs is that they allow potentially unlimited customisation and personalisation, with error-free knowledge

of the object of interest. Examples of their use include the dependency of brain tissue perfusion estimates on tracer delay in computed tomography imaging [2], the creation of a digital population spanning several organs for the optimisation of imaging protocols in the context of myocardial perfusion SPECT [3], the study of the effect of noise in the validation of the kurtosis model for estimations of brain perfusions using diffusion-weighted imaging in MRI [4], and the comparison of complication probability when using different modes of radiotherapy on lung tissue [5]. To our knowledge, no digital phantoms exist of the coronary arteries and/or of the heart.

The use of cardiac DPs, both static and dynamic, for the assessment of imaging modalities has accelerated over the last decade [6], along with similarly conceived cardiac atlases [7,8], to which we return below. Here, we present a static DP that our group has created for the assessment of derived haemodynamic modalities. Our tool represents the main vessels composing the left and right coronary arterial trees, and we demonstrate its use in the validation of our process in (i) the 3D geometry reconstruction and (ii) our computation of virtual fractional flow reserve (vFFR), both of which are embedded in our software suite VIRTUheart$^{TM}$. The detailed workflow of the software is described elsewhere (see [9,10]). However, our method and the associated DP tool are readily transferable to other clinically oriented workflows.

Briefly, the VIRTUheart$^{TM}$ software takes as input a pair of coronary angiograms obtained from different viewing angles and allows users to define the vessel endpoints of the segment of interest (typically within the vicinity of a local stenosis in the vessel). The software then reconstructs the 3D geometry of the selected segment. This is then meshed and, after the application of appropriate boundary conditions, computational fluid dynamics (CFD) analysis is carried out using ANSYS Fluent$^{TM}$ software to obtain the vFFR, i.e., the predicted ratio between the pressure distal to the stenosis and the aortic pressure. This software has been validated in both a clinical setting [9] and using a physical, 3D-printed phantom [10]. In this work we report the use of a DP to perform a more thorough and controlled error propagation study in order to provide insight on the effect of view selection, stenosis severity, and stenosis eccentricity on the accuracy of the 3D geometry reconstruction and hence on the computation of vFFR.

Our digital phantom is a minimally parameterised description, based upon a Cartesian grid, of vessel lumens of the most clinically important coronary arteries, and it interacts with CFD transparently. Its intended purpose is not to provide a "passive" cardiac statistical shape model (SSM), a novel cardiac atlas or, indeed, cardiac bioinformatics [11]. Indeed, more extensively parameterised SSMs and cardiac atlases have been applied in several ways: directly to visualise coronary arteries in 3D (a form of reconstruction), to parameterize epicardial surface geometry (myocardial shape), and to quantify its physiological, pathophysiological, and dynamical variation (especially the consequences of an infarct) by a number of workers [7,12]. We return to this matter in the Discussion section.

## 2. Materials and Methods

We first consider the definition and implementation of our DP and, secondly, the process by which it is deployed to generate luminal geometries for CFD simulation and to study error propagation in the vFFR workflow.

### 2.1. 3D Digital Phantom Anatomy Definition

The digital coronary phantom is constructed by first defining the coordinates of the vessels' centrelines in 3D space and subsequently two values of the radius at each point on the centerline: the two values correspond to the vessel radius on two mutually orthogonal axes lying on the plane normal to the local centerline tangent; if the two values are the same, the vessel cross-section will result in a circle; otherwise, it will define an ellipse with the two values corresponding to the lengths of the semi-axes. The user-defined variation of the radius controls the vessel taper and the profile of any stenosis. A point cloud of uniform density is then generated to define the fluid domain.

The DP vessel centreline coordinates and radii were based on several sources, including textbooks of coronary anatomy [13] and consultation with experienced cardiologists in our group. Within this study, we do not consider interpersonal variability of coronary tree geometry; rather, we use an anatomy that is considered to be a population average, based on the cardiologists' experience. Further geometrical and anatomical verification involved comparisons of two-dimensional (2D) projections of the DP anatomy with our groups' database of angiograms obtained clinically (baseline anatomy). The 3D coordinates of the points comprising the phantom are provided in a text file in this article's supplemental data.

This baseline anatomy was used to analyse the image segmentation and vessel reconstruction workflow steps, for a stenosis located in either the left circumflex (LCX) or left anterior descending artery (LAD). For each stenosis location, we considered diameter reductions of 0% (healthy vessel), 40%, 60%, and 80%, with both concentric and eccentric stenoses, resulting in ten distinct conditions for each stenosis location. This range of stenosis severity is considered to be clinically representative. In the eccentric case, we selected two mutually orthogonal axes from the normal plane to the centerline and applied different diameter reductions on each axis. The 3D anatomy of the digital phantom is shown in Figure 1 (left).

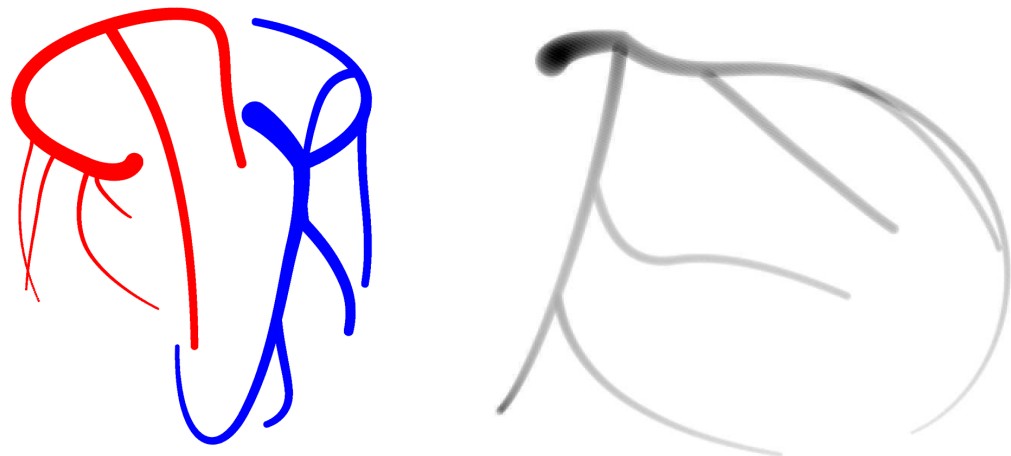

**Figure 1.** Digital phantom of the main vessels in the coronary arterial tree as respresented as a cloud of points (**left**), and an example of a derived, radiographically realistic projection (**right**). The left system is shown in blue and appears on the right side of the image; the right system is shown in red. This phantom represents the case in which no vessel presents stenosis.

### 2.2. 2D Image Generation Process

The 3D phantom was rotated and projected to create virtual angiograms, i.e., two-dimensional (2D) images, corresponding to the views taken following rotation of the angiography X-ray machine C-arm, which are typically defined in terms of the LAO/RAO angle, cranial/caudal angle, and the distance between the isocentre and the X-ray source and detector. The views are made radiographically realistic by computing the grayscale intensity of the 2D image from the number of voxels through which the virtual ray from the source to the detector travels. A representative projection, exhibiting realistic differential opacity, of the 3D anatomy is shown in Figure 1 (right). Finally, to provide representative noise, a stylised angiogram background based upon typical, representative angiogram features was developed in Adobe Photoshop™ and added to the DP views. The viewing angles of the background images were selected to match the views detailed in Table 1.

**Table 1.** Description of views used for digital angiograms.

| View Number | LAO | RAO | Cran | Caud | LF | LSD | Description |
|---|---|---|---|---|---|---|---|
| 1 | 0 | - | - | 31 | 765 | 1140 | Caud |
| 2 | - | 33 | - | 23 | 765 | 960 | RAO/Caud |
| 3 | - | 31 | 35 | - | 765 | 950 | RAO/Cran |
| 4 | 1 | - | 29 | - | 765 | 1150 | Cran |
| 5 | 24 | - | 34 | - | 765 | 1140 | LAO/Cran |
| 6 | 34 | - | - | 34 | 765 | 1193 | Spider |

For each stenosis location and degree, the phantom was projected onto three pairs of viewing angles which correspond to those typically used in the catheterisation laboratory to visualise stenoses in these locations. With reference to Table 1, the image pairs used to reconstruct the LCX were obtained using views 1 and 2, 1 and 6, and 2 and 6, while for the LAD, views 3 and 4, 3 and 5, and 4 and 5 were used. This resulted in 120 unique 2D images and 120 image pairs used for segmentation and 3D vessel reconstruction. Representative "virtual angiograms" are shown in Figures 2 and 3.

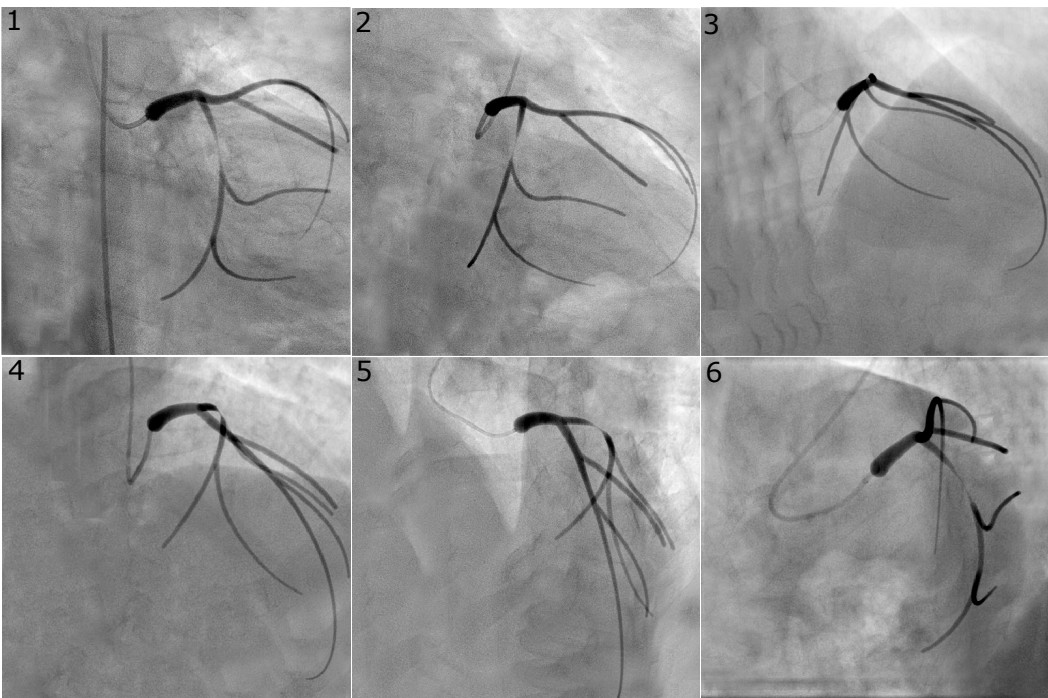

**Figure 2.** Virtual angiogram for all views of the left coronary system with a concentric stenosis of 80% diameter reduction in the LCX. Views **1**, **2**, and **6** were used to reconstruct this vessel as they are usually selected in the catheterisation laboratory since they avoid vessel overlap and excessive foreshortening for the vessel of interest.

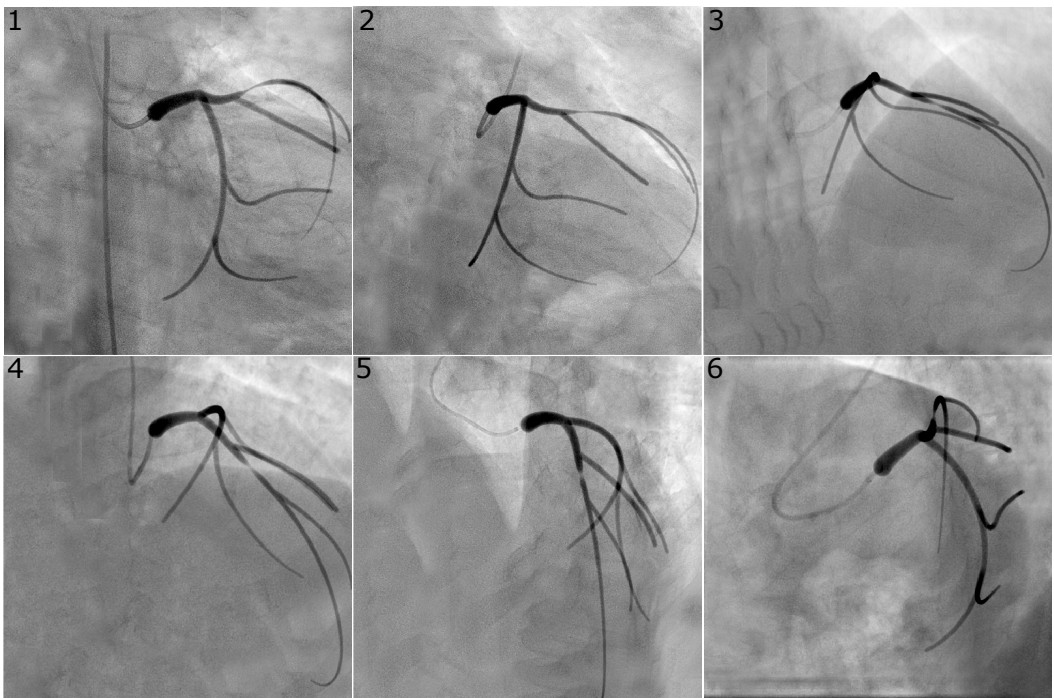

**Figure 3.** Virtual angiogram for all views of the left coronary system with an eccentric stenosis of type "8040" in the LAD. Views **3**, **4**, and **5** were used to reconstruct this vessel as they are usually selected in the catheterisation laboratory since they avoid vessel overlap and excessive foreshortening for the vessel of interest.

### 2.3. Image Segmentation, Reconstruction, and vFFR Computation

The resulting 120 "virtual angiogram" pairs were then processed using the VIRTUheart$^{TM}$ software, as explained in Section 1. The 3D arterial geometry was reconstructed from a pair of angiographic projections, meshed in ANSYS Fluent$^{TM}$, and appropriate boundary conditions were assigned for CFD analysis to compute the vFFR. All the CFD simulations solved incompressible Navier–Stokes equations using ANSYS Fluent$^{TM}$ 2021 R1, as described in [14]. The turbulence model was set to laminar, as our expectation was that the Reynolds numbers would not cross 500, which is still accepted as being in the laminar regime [15].

The benefit of the DP approach of this study is that the coronary anatomy used to create the phantom point cloud (and, hence, the 2D image projections) is known exactly (as it is mathematically defined). This allows a direct comparison to be made at the two stages of the VIRTUheart workflow, with (i) the results obtained using the digital phantom three-dimensional geometry (3D-DP) directly (no reconstruction involved) and (ii) the 3D geometry reconstructed from segmentation of the 2D "virtual angiogram" pairs (3D-IP).

An assessment of the accuracy of 3D geometry reconstruction was made by computing the following metrics (see Tables 2 and 3):

1.  The Hausdorff distance between 3D-DP and the 3D-IP centrelines;
2.  The absolute error between the 3D-DP and the 3D-IP mean radius;
3.  The norm of the difference between the pointwise radii across the common section of the vessel between the 3D-DP and the 3D-IP reconstructions;
4.  The 3D-DP and the 3D-IP stenotic radii.

**Table 2.** Assessment metrics for the LCX cases considered. The Hausdorff distance is that computed between the reconstructed artery and the digital phantom centrelines. The stenotic radii for the digital phantom (column $R_{sten}(DP)$) and the reconstruction (column $R_{sten}(Recon)$) are recoreded in mm.

| | Hausdorff | abs R | norm R | $R_{sten}$ (DP) | $R_{sten}$ (Recon) | abs $R_{sten}$ | rel $R_{sten}$ |
|---|---|---|---|---|---|---|---|
| **LCX0102** | | | | | | | |
| 0 | 0.15 | 0.08 | 0.11 | 1.31 | 1.16 | −0.15 | −12 |
| 4040 | 0.14 | 0.07 | 0.10 | 0.79 | 0.76 | −0.03 | −3 |
| 4060 | 0.16 | 0.05 | 0.08 | 0.64 | 0.64 | 0.00 | 0 |
| 4080 | 0.16 | 0.03 | 0.09 | 0.45 | 0.52 | 0.07 | 15 |
| 6040 | 0.16 | 0.05 | 0.08 | 0.64 | 0.67 | 0.02 | 4 |
| 6060 | 0.17 | 0.05 | 0.11 | 0.52 | 0.61 | 0.08 | 16 |
| 6080 | 0.16 | 0.03 | 0.09 | 0.37 | 0.46 | 0.09 | 24 |
| 8040 | 0.17 | 0.03 | 0.10 | 0.45 | 0.56 | 0.11 | 23 |
| 8060 | 0.18 | 0.02 | 0.10 | 0.37 | 0.49 | 0.12 | 32 |
| 8080 | 0.18 | 0.03 | 0.10 | 0.26 | 0.34 | 0.08 | 31 |
| **LCX0106** | | | | | | | |
| 0 | 0.26 | 0.02 | 0.07 | 1.31 | 1.27 | −0.04 | −3 |
| 4040 | 0.28 | 0.01 | 0.05 | 0.79 | 0.77 | −0.02 | −2 |
| 4060 | 0.28 | 0.03 | 0.09 | 0.64 | 0.80 | 0.15 | 24 |
| 4080 | 0.24 | 0.09 | 0.11 | 0.45 | 0.65 | 0.20 | 44 |
| 6040 | 0.28 | 0.02 | 0.06 | 0.64 | 0.69 | 0.05 | 8 |
| 6060 | 0.28 | 0.03 | 0.06 | 0.52 | 0.50 | −0.02 | −4 |
| 6080 | 0.28 | 0.09 | 0.11 | 0.37 | 0.49 | 0.12 | 32 |
| 8040 | 0.28 | 0.04 | 0.08 | 0.45 | 0.51 | 0.06 | 13 |
| 8060 | 0.27 | 0.04 | 0.07 | 0.37 | 0.42 | 0.05 | 13 |
| 8080 | 0.29 | 0.05 | 0.10 | 0.26 | 0.31 | 0.05 | 18 |
| **LCX0206** | | | | | | | |
| 0 | 0.26 | 0.04 | 0.07 | 1.31 | 1.35 | 0.04 | 3 |
| 4040 | 0.27 | 0.03 | 0.08 | 0.79 | 0.78 | −0.01 | −1 |
| 4060 | 0.29 | 0.03 | 0.07 | 0.64 | 0.62 | −0.03 | −4 |
| 4080 | 0.28 | 0.12 | 0.14 | 0.45 | 0.52 | 0.07 | 15 |
| 6040 | 0.26 | 0.03 | 0.07 | 0.64 | 0.60 | −0.04 | −6 |
| 6060 | 0.28 | 0.04 | 0.10 | 0.52 | 0.51 | −0.01 | −3 |
| 6080 | 0.29 | 0.11 | 0.17 | 0.37 | 0.42 | 0.04 | 12 |
| 8040 | 0.26 | 0.02 | 0.08 | 0.45 | 0.46 | 0.01 | 2 |
| 8060 | 0.27 | 0.10 | 0.13 | 0.37 | 0.42 | 0.05 | 13 |
| 8080 | 0.27 | 0.08 | 0.13 | 0.26 | 0.22 | −0.05 | −18 |
| 0–9% | 10–19% | > 20% | | | | | |

**Table 3.** Assessment metrics for the LAD cases considered. The Hausdorff distance is computed between the centerlines of the reconstructed artery and the digital phantom. The stenotic radii for the digital phantom (column $R_{sten}(DP)$) and the reconstruction (column $R_{sten}(Recon)$) are recorded in mm.

| | Hausdorff | abs R | norm R | $R_{sten}$ (DP) | $R_{sten}$ (Recon) | abs $R_{sten}$ | rel $R_{sten}$ |
|---|---|---|---|---|---|---|---|
| **LAD0304** | | | | | | | |
| 0 | 0.73 | 0.29 | 0.34 | 1.27 | 1.42 | 0.15 | 12 |
| 4040 | 0.72 | 0.28 | 0.34 | 0.77 | 0.76 | −0.01 | −1 |
| 4060 | 0.82 | 0.26 | 0.34 | 0.63 | 0.65 | 0.02 | 3 |
| 4080 | 0.73 | 0.27 | 0.34 | 0.44 | 0.55 | 0.11 | 24 |
| 6040 | 0.76 | 0.31 | 0.37 | 0.63 | 0.73 | 0.11 | 17 |
| 6060 | 0.72 | 0.28 | 0.34 | 0.51 | 0.57 | 0.06 | 12 |

**Table 3.** *Cont.*

| | Hausdorff | abs R | norm R | $R_{sten}$ (DP) | $R_{sten}$ (Recon) | abs $R_{sten}$ | rel $R_{sten}$ |
|---|---|---|---|---|---|---|---|
| 6080 | 0.84 | 0.27 | 0.34 | 0.36 | 0.39 | 0.03 | 9 |
| 8040 | 0.75 | 0.29 | 0.34 | 0.44 | 0.57 | 0.13 | 30 |
| 8060 | 0.85 | 0.27 | 0.35 | 0.36 | 0.45 | 0.09 | 24 |
| 8080 | 0.72 | 0.27 | 0.35 | 0.26 | 0.28 | 0.02 | 10 |
| **LAD0305** | | | | | | | |
| 0 | 0.76 | 0.23 | 0.33 | 1.27 | 1.38 | 0.11 | 8 |
| 4040 | 0.75 | 0.30 | 0.38 | 0.77 | 0.78 | 0.01 | 2 |
| 4060 | 0.75 | 0.30 | 0.37 | 0.63 | 0.63 | 0.01 | 1 |
| 4080 | 0.80 | 0.29 | 0.36 | 0.44 | 0.52 | 0.08 | 18 |
| 6040 | 0.74 | 0.31 | 0.37 | 0.63 | 0.79 | 0.16 | 26 |
| 6060 | 0.75 | 0.29 | 0.37 | 0.51 | 0.58 | 0.06 | 13 |
| 6080 | 0.77 | 0.30 | 0.36 | 0.36 | 0.39 | 0.03 | 8 |
| 8040 | 0.75 | 0.30 | 0.37 | 0.44 | 0.65 | 0.20 | 46 |
| 8060 | 0.77 | 0.30 | 0.37 | 0.36 | 0.49 | 0.12 | 34 |
| 8080 | 0.72 | 0.28 | 0.38 | 0.26 | 0.21 | −0.05 | −19 |
| **LAD0405** | | | | | | | |
| 0 | 0.72 | 0.11 | 0.21 | 1.27 | 1.28 | 0.00 | 0 |
| 4040 | 0.66 | 0.19 | 0.24 | 0.77 | 0.80 | 0.04 | 5 |
| 4060 | 0.62 | 0.21 | 0.26 | 0.63 | 0.60 | −0.03 | −5 |
| 4080 | 0.74 | 0.18 | 0.24 | 0.44 | 0.44 | −0.01 | −2 |
| 6040 | 0.64 | 0.20 | 0.25 | 0.63 | 0.77 | 0.15 | 24 |
| 6060 | 0.69 | 0.18 | 0.23 | 0.51 | 0.54 | 0.03 | 6 |
| 6080 | 0.65 | 0.18 | 0.23 | 0.36 | 0.37 | 0.01 | 2 |
| 8040 | 0.71 | 0.20 | 0.25 | 0.44 | 0.69 | 0.24 | 55 |
| 8060 | 0.72 | 0.20 | 0.25 | 0.36 | 0.51 | 0.15 | 41 |
| 8080 | 0.68 | 0.19 | 0.23 | 0.26 | 0.29 | 0.04 | 15 |

| 0–9% | 10–19% | > 20% | |
|---|---|---|---|

The geometries were then meshed to obtain a suitable input for CFD analysis. The optimum mesh was determined using Richardson's extrapolation [16], whereby mesh independence was checked and confirmed by calculating the predicted error in the simulation using three different mesh densities (see Appendix A).

To assess the accuracy of the vFFR computation, the latter was computed using both the 3D-DP and the 3D-IP geometries, using the same approach in both cases. To calculate the vFFR, we first estimated the maximal flow expected to pass through the artery under hyperaemic conditions. We estimated the flow, $Q_a$, through the stenosis, assuming an aortic pressure of $P_a = 100$ mmHg as an upper bound of expected aortic pressure, and used the Bernoulli principle to estimate the pressure loss across the stenosis [17]:

$$Q_a = \frac{-R_\mu + \sqrt{R_\mu^2 + 2\rho P_a(1 - f^2)/f^2 A_a^2}}{\rho(1 - f^2)/f^2 A_a^2}, \tag{1}$$

where $R_\mu$ is the microvascular resistance in the hyperaemic state, $\rho = 1056$ kg/m$^3$ is the density of blood, f is the ratio of the cross-sectional area at the throat of the stenosis to that of the inlet, and $A_a$ is the area of the inlet.

We then characterised the pressure drop vs. flow relationship for each 3D arterial geometry by running two steady-state simulations, the first with inlet flow $Q_1 = q_a$ and the second with inlet flow $Q_2 = q_a/3$. Distal pressure was set to 0Pa for both analyses. The fluid was modelled as a viscous fluid with density $\rho = 1056$ kg/m$^3$ and viscosity $\nu = 0.0035$ Pa s. For each simulation, distal pressure $P_i$ was computed at the point where a clinician is expected to position the pressure catheter. The pressure drop was then computed by $\Delta p = P_a - P_i$ for both simulations and used to calculate the characterisation parameters according to [17]:

$$z_2 = \frac{\Delta p_1 Q_2 - \Delta p_2 Q_1}{Q_1^2 Q_2 - Q_1 Q_2^2}, \tag{2}$$

$$z_1 = \frac{-\Delta p_1 Q_2^2 + \Delta p_2 Q_1^2}{Q_1^2 Q_2 - Q_1 Q_2^2}. \tag{3}$$

To calculate the vFFR, we assumed an average pressure at the inlet of $p = 85$ mmHg, which is a value expected in most clinical cases. We calculated the flows through the artery at baseline ($Q_b$) and hyperaemia ($Q_h$) by varying the resistance $R$ between hyperaemic ($R_h = 8.72 \times 10^9$ Pa s m$^{-3}$) and baseline ($R_b = 1.55 \times 10^{10}$ Pa s m$^{-3}$) values, from Equation (4):

$$Q = \frac{-(z_1 + R) + \sqrt{(z_1 + R)^2 + 4z_2 p}}{2z_2}. \tag{4}$$

Finally, the vFFR value was calculated as the pressure in the microvasculature according to Equation (5):

$$\text{vFFR} = \frac{R \, Q}{p}, \tag{5}$$

where $R, Q$ took values $R_b, Q_b$, and $R_h, Q_h$, respectively, for the baseline and hyperaemic states, respectively. vFFR values were compared between those obtained using the 3D-DP geometry and those obtained using each of the three 3D-IP geometries.

## 3. Results

In this section, the following naming convention is used; virtual angiogram pairs are referenced by "LCX0102", where the first three characters denote the vessel of interest (LCX or LAD) and the numbers denote the pair of views used (1, 2). The stenosis configuration is referenced using four-digit case names such as "0000", "4040", or "4080" to denote the stenosis type and severity. "0000" is the healthy case; the same first two digits repeated twice, e.g., "4040", denote a concentric stenosis of that diameter reduction; a case with two distinct digits, such as "4080", represents an eccentric stenosis with each diameter reduction applied on orthogonal axes in the plane normal to the vessel centreline (thus, "4080" and "8040" represent the same total reduction in area, but with different orientations of eccentricity).

### 3.1. 2D Virtual Angiogram Generation

Figures 2 and 3 show representative examples of virtual angiograms generated for the two coronary arteries considered in this study. Figure 2 shows the case in which a concentric stenosis, of 80% diameter reduction, affects the LCX. This case is included here for theoretical interest as it allows for an appreciation of the severity of a stenosis of this diameter reduction, but it is of smaller concern to the clinical community, due to the certainty of its severity. Figure 3 shows the case in which the LAD is affected by stenosis, this time of the eccentric type, with diameter reductions of 80% and 40% in the two orthogonal directions. This case shows how eccentric stenoses can appear differently in different views, and showcases that the choice of the view pair can result in either underestimation or overestimation of the severity of the stenosis (thus, overestimating or underestimating the radius, respectively).

### 3.2. Accuracy of 3D Geometry Reconstruction

Tables 2 and 3 report, for the LCX and LAD geometries respectively, several metrics used to assess the accuracy of the 3D geometry reconstruction from virtual angiogram pairs relative to the 3D-DP geometry, as described in Section 2.3. In these tabulations we record the Hausdorff distance between the digital phantom centreline and that of its

reconstruction, the absolute error in mean radius, the norm of the difference in pointwise radii, and the stenotic radii in the digital phantom and the reconstruction.

An example of a mesh generated from a reconstruction is shown in Figure 4.

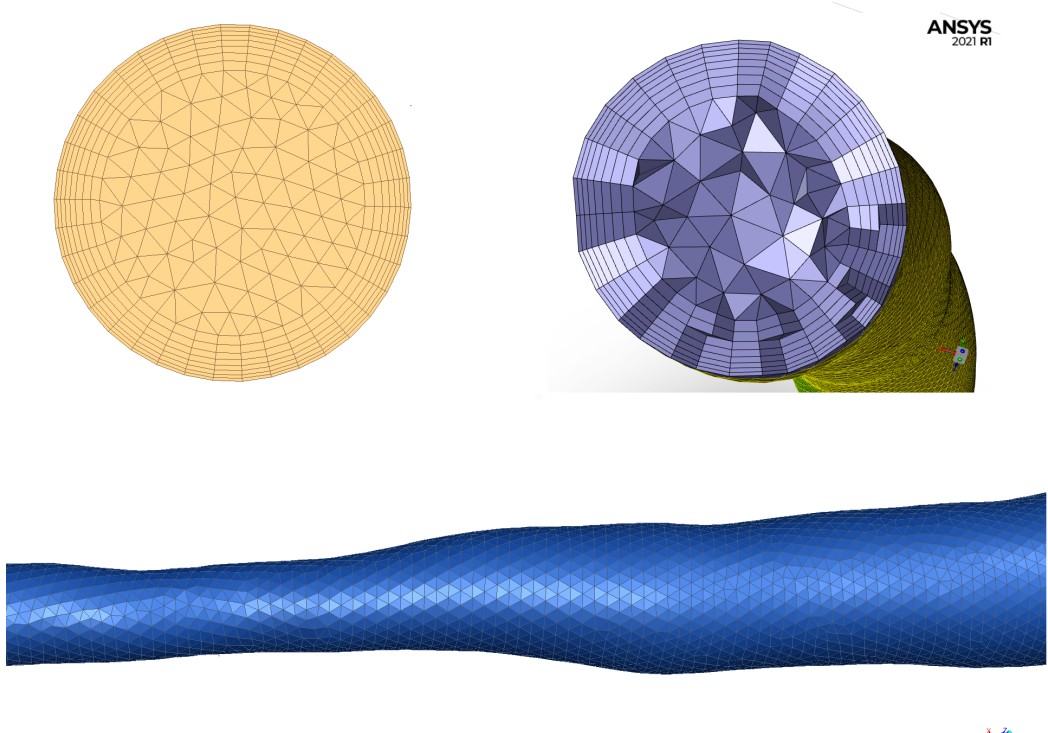

**Figure 4.** Example of mesh generated from a 3D reconstruction of case *LCX0102_4060*. **Top left corner**: inlet face, **top right**: cross-section in stenosed region, and **bottom**: outside wall.

### 3.3. Accuracy of vFFR Computation

Tables 4 and 5 report the values of vFFR, for both baseline and hyperaemia, obtained using the 3D-DP geometry, and the difference between these values and those computed using the reconstructions from each of the three pairs of virtual angiograms for both the LCX and LAD stenosis locations. Table 6 reports the absolute error in the computation of vFFR and the relative error in the estimation of the radius at the point of maximum stenosis.

**Table 4.** Virtual FFR values at baseline (B) and hyperaemia (H) for the left circumflex (LCX) model and the original digital phantom (DP), the reconstruction using views 1 and 2 (LCX0102), 1 and 6 (LCX0106), and finally, 2 and 6 (LCX0206).

|  | 3D-DP | | LCX0102 | | LCX0106 | | LCX0206 | |
| --- | --- | --- | --- | --- | --- | --- | --- | --- |
|  | **B** | **H** | **B** | **H** | **B** | **H** | **B** | **H** |
| 0000 | 0.94 | 0.88 | 0 | 0 | 0.01 | 0.02 | 0.02 | 0.04 |
| 4040 | 0.92 | 0.86 | 0 | −0.01 | 0.01 | 0.01 | 0.03 | 0.04 |
| 4060 | 0.91 | 0.83 | 0 | 0 | 0.02 | 0.04 | 0.02 | 0.04 |
| 4080 | 0.82 | 0.69 | 0.06 | 0.09 | 0.11 | 0.18 | 0.11 | 0.16 |
| 6040 | 0.91 | 0.83 | 0 | 0.01 | 0.02 | 0.03 | 0.02 | 0.04 |
| 6060 | 0.88 | 0.77 | 0.02 | 0.04 | 0.01 | 0.02 | 0.02 | 0.03 |
| 6080 | 0.75 | 0.59 | 0.1 | 0.13 | 0.14 | 0.2 | 0.07 | 0.09 |
| 8040 | 0.82 | 0.69 | 0.07 | 0.11 | 0.07 | 0.1 | 0.05 | 0.07 |
| 8060 | 0.75 | 0.59 | 0.11 | 0.16 | 0.08 | 0.1 | 0.09 | 0.11 |
| 8080 | 0.53 | 0.36 | 0.2 | 0.21 | 0.12 | 0.12 | −0.12 | −0.09 |

| 0–0.02 | 0.03–0.06 | > 0.06 |
| --- | --- | --- |

**Table 5.** Virtual FFR values at baseline (B) and hyperaemia (H) for the left anterior descending (LAD) artery and the original digital phantom (DP), the reconstruction using views 3 and 4 (LAD0304), 3 and 5 (LAD0305), and finally, 4 and 5 (LAD0405). The † symbol indicates cases for which the pressure converged, but the residual for the continuity equation remained higher than the convergence threshold.

| | 3D DP | | LAD0304 | | LAD0305 | | LAD0405 | |
|---|---|---|---|---|---|---|---|---|
| | **B** | **H** | **B** | **H** | **B** | **H** | **B** | **H** |
| 0000 | 0.97 | 0.94 | 0.02 | 0.03 | 0.01 | 0.02 | −0.01 | −0.04 |
| 4040 | 0.96 | 0.92 | 0.01 | 0.02 | 0.01 | 0.02 | 0.01 | 0.02 |
| 4060 | 0.95 | 0.90 | 0.01 | 0.01 | 0.01 | 0.01 | 0 | −0.01 |
| 4080 | 0.88 | 0.77 | 0.06 | 0.09 | 0.05 | 0.09 | 0 | −0.01 |
| 6040 | 0.95 | 0.90 | 0.02 | 0.04 | 0.02 | 0.05 | 0.01 | 0.02 |
| 6060 | 0.93 | 0.85 | 0.02 | 0.03 | 0.01 | 0.01 | 0.01 | 0.01 |
| 6080 | 0.81 | 0.66 | 0.01† | 0.01† | 0.04 | 0.04 | 0.03 | 0.04 |
| 8040 | 0.88 | 0.77 | 0.07 | 0.11 | 0.08 | 0.15 | 0.07 | 0.12 |
| 8060 | 0.81 | 0.66 | 0.08 | 0.11 | 0.11 | 0.17 | 0.1 | 0.16 |
| 8080 | 0.59† | 0.41† | 0.04† | 0.04† | −0.18† | −0.15† | 0.13† | 0.13† |

| 0–0.02 | 0.03–0.06 | > 0.06 |
|---|---|---|

**Table 6.** Summary of absolute FFR error and relative stenosis radius error for all cases.

| | | Change from 3D-DP | | | LCX % Error Stenosis Radius | | |
|---|---|---|---|---|---|---|---|
| | 3D-DP | LCX0102 | LCX0106 | LCX0206 | LCX0102 | LCX0106 | LCX0206 |
| 0 | 0.88 | 0 | 0.02 | 0.04 | −12 | −3 | 3 |
| 4040 | 0.86 | −0.01 | 0.01 | 0.04 | −3 | −2 | −1 |
| 4060 | 0.83 | 0 | 0.04 | 0.04 | 0 | 24 | −4 |
| 4080 | 0.69 | 0.09 | 0.18 | 0.16 | 15 | 44 | 15 |
| 6040 | 0.83 | 0.01 | 0.03 | 0.04 | 4 | 8 | −6 |
| 6060 | 0.77 | 0.04 | 0.02 | 0.03 | 16 | −4 | −3 |
| 6080 | 0.59 | 0.13 | 0.2 | 0.09 | 24 | 32 | 12 |
| 8040 | 0.69 | 0.11 | 0.1 | 0.07 | 23 | 13 | 2 |
| 8060 | 0.59 | 0.16 | 0.1 | 0.11 | 32 | 13 | 13 |
| 8080 | 0.36 | 0.21 | 0.12 | −0.09 | 31 | 18 | −18 |
| | | Change from 3D-DP | | | LAD % Error Stenosis Radius | | |
| | 3D-DP | LAD0304 | LAD0305 | LAD0405 | LAD0304 | LAD0305 | LAD0405 |
| 0 | 0.94 | 0.03 | 0.02 | −0.04 | 12 | 8 | 0 |
| 4040 | 0.92 | 0.02 | 0.02 | 0.02 | −1 | 2 | 5 |
| 4060 | 0.9 | 0.01 | 0.01 | −0.01 | 3 | 1 | −5 |
| 4080 | 0.77 | 0.09 | 0.09 | −0.01 | 24 | 18 | −2 |
| 6040 | 0.9 | 0.04 | 0.05 | 0.02 | 17 | 26 | 24 |
| 6060 | 0.85 | 0.03 | 0.01 | 0.01 | 12 | 13 | 6 |
| 6080 | 0.66 | 0.01 | 0.04 | 0.04 | 9 | 8 | 2 |
| 8040 | 0.77 | 0.11 | 0.15 | 0.12 | 30 | 46 | 55 |
| 8060 | 0.66 | 0.11 | 0.17 | 0.16 | 24 | 34 | 41 |
| 8080 | 0.41 | 0.04 | −0.15 | 0.13 | 10 | −19 | 15 |

| | vFFR Error Bands | | | % Stenosis Radius Error Bands | | |
|---|---|---|---|---|---|---|
| | 0–0.02 | 0.03–0.06 | >0.06 | 0–9% | 10–19% | > 20% |
| | Disagreement in threshold level | | | | | |

It should be highlighted that five cases in the LAD set had convergence issues, i.e., the pressure converged but the residual of the continuity equation remained higher than the convergence threshold. In particular, the latter displayed oscillatory behaviour, which suggests that perhaps a steady-state solution does not exist. Four out of five of these cases are the concentric stenosis with 80% diameter reduction: this corresponds to a 96% area reduction and is thus outside clinical interest and outside the scope of our software. The fifth case is a "6080", which represents a 92% area reduction; therefore the same reasoning applies. Indeed, our selection of vessels has been dictated by theoretical interest as well as

clinical need. The matter of computational tractability in the resulting numerical problem is an independent question: if one generates a problematic geometry digitally (for which a steady-state solution does not exist, say), one must expect associated challenges with numerical convergence.

## 4. Discussion

In this study, we have developed a DP to allow the assessment of the accuracy of 3D geometry reconstruction and the subsequent computation of vFFR, using the VIRTUheart$^{TM}$ software framework. Although we present a specific instance of use of this digital phantom as an exemplar, this approach could be used in any other software framework dealing with medical image processing and/or CFD analysis on arterial geometries. We have considered the implications of imaging stenoses located in both the LCX and LAD vessels, using several image pairs in each case, with a stenosis severity that represents (and is indeed larger than) the range expected to be observed in clinical practice.

A comparison of the accuracy of the VIRTUheart$^{TM}$ vessel reconstructions is reported in Tables 2 and 3 of the Results section. These data demonstrate the overall accuracy of the reconstruction of the vessel centreline, with the magnitude of the Hausdorff distance between 3D-DP and 3D-IP reconstructions being less than 1 mm in all cases. The Hausdorff distance was less for the LCX (all values < 0.3 mm) compared with the LAD (all values < 0.85 mm). The ability of the 3D-IP reconstruction to capture the radius of the vessel was good for LCX reconstruction, with the error in absolute average value being < 0.12 mm in all cases, and the $L^2$-norm of the pointwise difference along the vessel length being < 0.17 mm. The error increased for the reconstruction of the LAD, with an error in absolute average value of < 0.31 mm and a $L^2$-norm of the pointwise difference of < 0.38 mm. The reduced accuracy in LAD reconstruction compared to the LCX is most likely due to the foreshortening of the former vessel at its distal end in all views. Indeed, while it was possible to reconstruct the LCX artery almost entirely, i.e., with the distal endpoint being very close to the endpoint of the vessel, in the case of the LAD, the distal endpoint of the segmentation had to be selected a small distance before the actual vessel endpoint due to significant foreshortening. When computing the Hausdorff distance, the two centerlines (3D-DP and 3D-IP) are registered so that either the distal or the proximal end coincide. The larger discrepancy between 3D-DP and 3D-IP endpoints for the LAD artery then results in a larger Hausdorff distance between the two centrelines.

There was relatively little variation in any of the error metrics with choice of image pairs or with stenosis severity, as they all represent the ability of the reconstruction to capture the overall form of vessel anatomy, rather than the detail of the stenosis itself. As a result, these error metrics are expected to have small influence on the computed vFFR values.

The accuracy of the reconstruction of the vessel radius at the stenosis location was expected to be closely related to the accuracy of the computed vFFR. Greater variation of this metric was observed with choice of image pairs with a range of maximum absolute error from 0.07 mm to 0.20 mm for the LCX and 0.15 mm to 0.24 mm for the LAD. There was no clear relation between absolute error and the stenosis severity; as a result, the percentage error in the stenosis radius tended to increase for larger-percentage diameter reductions. The error in stenosis reconstruction was typically larger for cases with eccentric stenosis, as expected, due to the different views showing different vessel radii depending on the orientation of the eccentricity. Indeed, as visible in Figure 3, which reports a case of eccentric stenosis in the LAD, the vessel radius appears significantly larger in views 4 and 5, compared to, for example, views 1 and 2. Our software estimates the vessel radius from each of the two angiographic views and averages the two values; the reconstructed local lumen will have a circular shape with a radius equal to that averaged value. It is possible for two views to simultaneously fail to capture the peak value of stenosis, thus obtaining a larger error in the radius estimation and therefore in the vFFR.

A comparison of the vFFR values computed using the 3D-DP geometry and the 3D-IP geometry is reported in Tables 4 and 5. This data demonstrates similar outcomes when reconstructing stenoses in both the LCX and LAD vessels. For the cases 0000, 4040, 4060, and 6040, the error in the computed vFFR is <0.05, and all vFFR values are above the typical threshold of 0.8 that is used to determine if intervention is required. For the 6060 case, the error in the computed vFFR remains less than 0.05, and for the LAD, the vFFR values remain above the threshold for intervention. For the LCX, the hyperaemic vFFR computed using the 3D-DP geometry is 0.77, and the error in vFFR is sufficient to raise this above 0.8 for image pairs 0102 and 0206, but not for image pair 0106. The error becomes more pronounced for all cases involving an 80% diameter reduction on at least one axis of the stenosis. The only exception to this is the 6080 case in the LAD, where all vFFR errors < 0.05 and all values agree in terms of relation to the 0.8 threshold. This increased error results in disagreement between the 3D-DP data and the 3D-IP data, in terms of the relation to the 0.8 threshold for the 4080, 8040, and 8060 cases across both the baseline and hyperaemic simulations. It is notable that there is variation in agreement with the choice of image pair used. For the 8080 case, although the error in vFFR is large, there is no disagreement between the 3D-DP and 3D-IP values in terms of their relation to the threshold, due to the low values of vFFR for all image pairs.

Less reassuringly, this study also shows that care must be taken in the catheterisation laboratory when selecting an optimal view that highlights the stenosis. Indeed, especially in the case of eccentric stenosis, it is possible for both views to not show maximum stenosis and, thus, overestimate the FFR, in turn underestimating the severity of the disease.

Our software encountered some difficulties when processing cases with very severe stenoses. All cases of 92% or 96% area reduction (i.e., "6080", "8060", and "8080") required increased contrast between the phantom projection and the added background in the digital angiographies. This was due to the fact that the number of voxels traversed by the light (and, thus, the resulting shade of gray in the projection) in the area of peak stenosis was very low; thus, the vessel (the diameter of which spanned 1 to 2 pixels in the projection) was indistinguishable from the background to our image processing algorithm. This problem was overcome by slightly increasing the contrast between the projected phantom vessel and the background.

The CFD analysis also encountered convergence issues for all cases of 96% and one case of 92% area reduction. Although the pressure converged, the residual in the continuity equation displayed non-convergent oscillatory behaviour that might indicate the absence of a steady-state solution. This is not a consequence of using the DP methodology, and both the above issues are not unexpected. We have indeed considered a range of stenosis severity that exceeds that of clinical significance, but remains of interest from the theoretical perspective of CFD analysis. Our software is designed to deal with cases that are of clinical interest and, in particular, to help inform treatment decisions for cases that are close to the treatment decision boundary, which is generally taken at an FFR value of 0.8. Accordingly, cases of such high vessel diameter reduction are outside the scope of our software and, in real life, are likely to be severe enough that treatment is deemed necessary, with all estimations or measurements of FFR considered to be superfluous.

Fractional flow reserve is the gold-standard method of assessing lesion severity in coronary artery disease and is recommended in international guidelines. However, due to the increased cost and patient time on the table, its use is not yet as widespread as would be desirable [18]. In recent years, several software suites that aim to estimate FFR from coronary angiographies have emerged [9,19–21]. All software packages were validated against invasive measurements of FFR via pressure wires, and one was additionally validated against an in vitro model of the coronary circulation [19]. A systematic review and meta-analysis of the clinical validation studies confirms the good accuracy of the software packages, with small differences between them [22], although no comparison has been made to date on a shared set of cases. A variation of this software that computes absolute flow within a coronary artery has been validated both clinically and against an in vitro

model [23]. The methodology of the digital phantom could be adopted by any of these software packages, not only as a method for software validation, but also as a tool to identify sources of error to address with software improvements. The widespread adoption of such software has the potential to significantly improve patient care in the context of coronary artery disease, as it bypasses the obstacles presented by the invasive measurement of FFR [24]. For this reason, it is highly desirable that these softwares continually improve their accuracy and computational efficiency: the digital phantom could be a valuable tool in facilitating this progress.

Recent years have seen growing interest in the development of cardiac atlases [7,25], i.e., libraries of patient-specific medical images of cardiac anatomy and associated clinical data. These atlases can be used to inform statistical shape models (SSMs), i.e., mathematically defined geometries parametrised so as to allow the spanning of population-wide possible geometries by varying shape-defining parameters. Some libraries include time-dependent data and are, therefore, also capable of generating $3D + t$ models, which allow the study of heart motion over the cardiac cycle [12,26]. SSMs are also valuable tools for the study of population-wide variability in cardiac anatomy, (patho-)physiology, and bioinformatics. For our purposes, however, these tools include complexity and parametrisation requirements that are outside the scope of this work. As we state from the outset, the objective of our DP is to create geometries that are mathematically defined and realistic, but computationally tractable with the minimum number of parameters that would allow reasonable customisation, without adding unnecessary complexity that would hinder the identifiability of error sources and modes of error propagation. However, their use is a possible research direction for our work, following the workflow of Catalano et al. in [27], for example. In the above, a statistical shape model of the aortic arch is constructed from a library of $n = 106$ patients, an average "template" shape is identified, and principal component analysis is performed to identify modes of variation of the geometry across the population. A set of representative geometries is then generated by varying each mode by 1.5 and 3 standard deviations, and finally, CFD analysis is performed on the generated set. The application of a similar workflow to coronary artery disease would validate the use of the digital phantom against real patient-specific data, and could potentially reveal the anatomical features that have the greatest impact on haemodynamic metrics of interest and, thus, help assess lesion severity.

The CFD analysis tool used in our software solves the Navier–Stokes equations as standard practice, but the application of the digital phantom presented in this work is not limited to this case. Indeed, it is defined on a Cartesian grid, which is the native coordinate system for a majority of medical image standards. The definition of the geometrically complex luminal boundary on such a grid is very convenient for more novel CFD tools, such as lattice Boltzmann simulations, which may have a larger role to play in CFD workflows in the future.

The DP presented in this work is generated as a cloud of points, and we attach the coordinates of the constitutive points to this manuscript for both the complete phantom and for individual vessels, with the hope that it can be used and shared in the community or motivate the use of similar methodologies.

## 5. Conclusions

Digital phantoms (DPs) are useful tools for computational modelling software that involves the analysis of patient-specific or population-wide geometry data, supporting accuracy estimation, error propagation studies, and the identification of error sources. We have presented the use of a DP for error propagation analysis using our in-house VIRTUheart™ software to evaluate the effect of stenosis severity, eccentricity, and view selection on: (a) the accuracy of 3D reconstruction, and (b) the accuracy of FFR estimation. The crucial advantages provided by a digital phantom are the broad possibilities for customisation and the definition of a ground truth 3D geometry which is mathematically defined. As a

result, the DP represents a fast and inexpensive validation method for use in pre-clinical validation stages.

**Author Contributions:** D.R.H., G.P. and P.D.M. devised the digital phantom; G.P., K.C. and N.G. deployed it, generated the data, and interpreted it; support was provided in the form of protocols developed by D.R.H., A.J.N., K.C. and I.H.; J.P.G. and P.D.M. devised the clinical vFFR workflow and the anatomical basis and parameterisation of the phantom for clinical application; G.P., K.C., A.J.N. and I.H. wrote the article. All authors have read and agreed to the published version of the manuscript.

**Funding:** This work has received funding from the European Union's Horizon 2020 research and innovation programme under grant agreement No. 857533: Sano Centre for Computational Medicine, Kraków, Poland (https://sano.science/, accessed on 9 June 2022), and from the British Heart Foundation "Virtu-4" grant (TG/19/1/34451). Nada Ghorab is funded by the EPSRC Doctoral Training Partnership (EP/T517835/1). Paul Morris was funded by the Wellcome Trust (214567/Z/18/Z). For the purpose of Open Access, the author has applied a CC BY public copyright licence to any Author-Accepted Manuscript version arising from this submission.

**Institutional Review Board Statement:** Not applicable.

**Informed Consent Statement:** Not applicable.

**Data Availability Statement:** Text files in VTK format are made available containing the 3D Cartesian coordinates ($x, y, z$ in the three columns) of the cloud points comprising the complete phantom of the left arterial tree ("complete"), as well as the individual vessels within it ("LCX" for the Circumflex, "LAD" for the left anterior descending artery, "Diag1" and "Diag2" for the diagonal branches of the LAD, "LM" for the left main, "LOM" for the obtuse marginal, and finally, "PLV" for the posterior LV branch).

**Conflicts of Interest:** The authors declare no conflict of interest.

## Abbreviations

The following abbreviations are used in this manuscript:

| | |
|---|---|
| CFD | Computational Fluid Dynamics |
| DP | Digital Phantom |
| FFR | Fractional Flow Reserve |
| LAD | Left Anterior Descending |
| LCX | Left Circumflex |
| SSM | Statistical Shape Model |
| vFFR | virtual FFR |

## Appendix A. Grid Independence Study

The grid independence study is performed on case *LCX*0102_4060 on three levels of refinement, where at each step, the number of elements is doubled (mesh 1 is coarsest, 3 is finest). The edge length for each level of refinement is reported in Table A1. The pressure value is extracted at the centre of the inlet. The mean error is 0.054% and the maximum error is 0.162%. Considering the error threshold of 1%, as indicated in [16], it can be concluded that our simulations are grid-independent. The middle value of mesh density is used throughout the work presented in this article.

**Table A1.** Grid independence study using Richardson's extrapolation. Meshes change from coarsest to finest with the numer of elements doubling at each step. Pressure values are extracted at the centre of the inlet.

| Mesh | Edge Length (mm) | Static Pressure (Pa) |
|:---:|:---:|:---:|
| 1 | 0.252 | 2892.683 |
| 2 | 0.200 | 2892.661 |
| 3 | 0.159 | 2888 |

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
