# Peer review of "The Use of Digital Coronary Phantoms for the Validation of Arterial Geometry Reconstruction and Computation of Virtual FFR"

_fluids, doi:10.3390/fluids7060201_

Round 1
Reviewer 1 Report
Topic: The use of digital coronary phantoms for the validation of arterial geometry reconstruction and computation of virtual FFR
The above manuscript has been reviewed by me. The following points needs to be incorporated for value addition in the manuscript:
- Sources of all equations needs to be given.
- Please add the governing equations used in CFD Modelling along with the name of models used for simulation.
- Please show mesh in the geometry used for CFD Simulation.
- Please include Grid Independent test.
- Please also add results validation from experimental results / simulation results of other researchers.
- Conclusion need to be rewritten and summarized.
- " et al." needs to be removed from references 6 and 8.
Reviewer 2 Report
The manuscript presented was very well cared for both in the expository part of the problematic and in the part of exposition of the values.
The application encourages development in this sector and the use of CFD is very important in the premonition of scenarios.
I have no notes to report and the article is already acceptable in its current form.
I thank the authors for the clarity of presentation and the commitment to deepen the theme.
Author Response
Dear reviewer,
Thank you very much for taking the time to evaluate our manuscript. We are glad if you found it interesting and join you in hoping that it will promote progress in this field.
Thank you again for you time and effort,
Giulia Pederzani